# SARS-CoV-2 genetic diversity in Venezuela: Predominance of D614G variants and analysis of one outbreak

Carmen L. Loureiro[1], Rossana C. Jaspe[1], Pierina D´Angelo[2], José L. Zambrano[3], Lieska Rodriguez[2], Víctor Alarcon[2], Mariangel Delgado[4], Marwan Aguilar[2], Domingo Garzaro[1], Héctor R. Rangel[1], Flor H. Pujol [1] *

1 Laboratorio de Virología Molecular, Centro de Microbiología y Biología Celular, Instituto Venezolano de Investigaciones Científicas, Caracas, Miranda, Venezuela, 2 Instituto Nacional de Higiene "Rafael Rangel", Caracas, Miranda, Venezuela, 3 Laboratorio de Biología de Virus, Centro de Microbiología y Biología Celular, Instituto Venezolano de Investigaciones Científicas, Caracas, Miranda, Venezuela, 4 Unidad Unidad de Microscopia Electrónica y Confocal, Centro de Microbiología y Biología Celular, Instituto Venezolano de Investigaciones Científicas, Caracas, Miranda, Venezuela

* fhpujol@gmail.com

**Data Availability Statement:** Nucleotide sequence data have been deposited into the GenBank database under the accession numbers

## Abstract

SARS-CoV-2 is the new coronavirus responsible for COVID-19 disease. The first two cases of COVID-19 were detected in Venezuela on March 13, 2020. The aim of this study was the genetic characterization of Venezuelan SARS-CoV-2 isolates. A total of 7 full SARS-CoV-2 genome sequences were obtained by Sanger sequencing, from patients of different regions of Venezuela, mainly from the beginning of the epidemic. Ten out of 11 isolates (6 complete genomes and 4 partial spike genomic regions) belonged to lineage B, bearing the D614G mutation in the Spike protein. Isolates from the first outbreak that occurred in the Margarita Island harbored an in-frame deletion in its sequence, without amino acids 83–85 of the NSP1 of the ORF1. The search for deletions in 48,635 sequences showed that the NSP1 gene exhibit the highest frequency of deletions along the whole genome. Structural analysis suggests a change in the N-terminal domain with the presence of this deletion. In contrast, isolates circulating later in this island lacked the deletion, suggesting new introductions to the island after this first outbreak. In conclusion, a high diversity of SARS-CoV-2 isolates were found circulating in Venezuela, with predominance of the D614G mutation. The first small outbreak in Margarita Island seemed to be associated with a strain carrying a small deletion in the NSP1 protein, but these isolates do not seem to be responsible for the larger outbreak which started in July.

## Introduction

On March 11, the World Health Organization declared a pandemic COVID-19, the disease caused by a new coronavirus which emerged in Wuhan at the end of 2019. Since then, at the end of January 2021, more than 100 million persons worldwide have contracted the disease, with more than 2,000,000 deaths [1]. Once originated in China, the epidemic center evolved

MT907515-MT907521, MW015946-MW015954 and MW040500-MW040503.

**Funding:** This study was supported by Ministerio del Poder Popular de Ciencia, Tecnología e Innovación of Venezuela.

**Competing interests:** The authors have declared that no competing interests exist.

first in Asia, then in Europe, and in August 2020, the epicenter was located in the Americas, including Latin America.

SARS-CoV-2, the new coronavirus responsible for COVID-19 disease, belongs to the family *Coronaviridae*, genus *Betacoronavirus*, and shares with its predecessor SARS-CoV the subgenera *Sarbecovirus*. After interaction of the Receptor Binding Domain (RBD) of the viral Spike protein with the viral receptor ACE2, SARS-CoV-2 enters the cell through a fusion mechanism. The viral fusion protein is exposed by priming with a proteolytic enzyme [2]. The viruses of this family possess the largest known continuous RNA genome (around 30,000 nt). The RNA-dependent RNA polymerase complex (RdRP) possess proofreading capacity, implying that the mutation rate of these viruses is lower than the one observed for other RNA viruses [3, 4]. On the other hand, these viruses display also a high frequency of recombination, horizontal gene transfer, gene duplication, and alternative open reading frames, which allow them to jump from one species to another [5].

The first case of COVID-19 was detected in Venezuela on March 13, 2020 (Fig 1). The first viral introductions were mainly through air flight passengers, mainly from Europe, during the month of March. Immediately after the two first cases, quarantine and the use of face mask was mandatory. The second wave of viral introductions started in April and includes migrants returning from neighboring countries, mainly Colombia and Brazil. The aim of this study was the genetic characterization of Venezuelan SARS-CoV-2 isolates and the analysis of the ongoing epidemic in two settings: the Margarita Island and the second most populous city, Maracaibo.

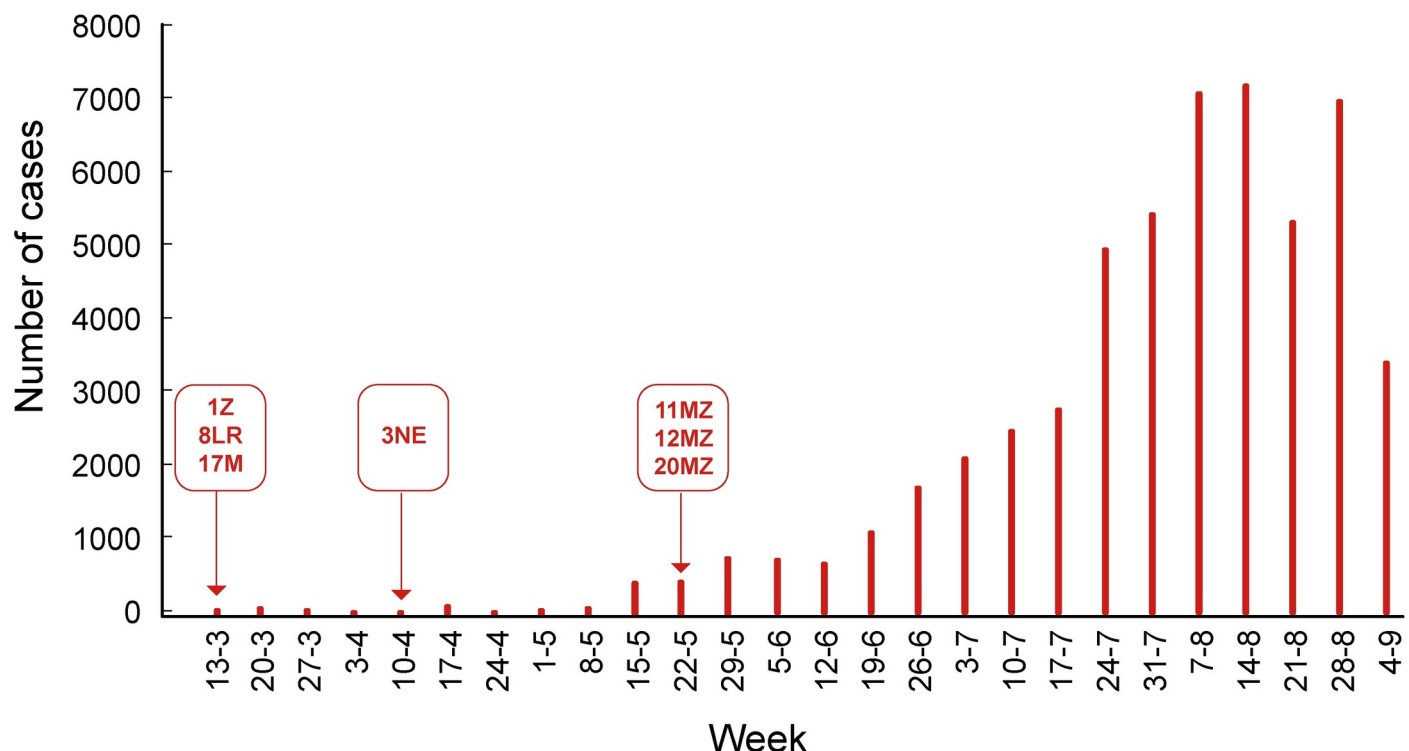

**Fig 1. COVID-19 cases in Venezuela, according to official data.** The cases are referred to as the date of report and downloaded from the web page http://www.ucv.ve/cendes. The arrows indicate the date of collection of the samples for which complete genome sequences were obtained.

## Materials and methods

### Amplification and sequencing

Clinical samples were processed at the Instituto Nacional de Higiene "Rafael Rangel", as part of the national diagnostic response to the COVID-19 pandemic, and sent as anonymous samples to Laboratorio de Virología Molecular, Instituto Venezolano de Investigaciones Científicas (IVIC), for sequencing. The samples were chosen according to the following criteria: from the beginning of the epidemic (for complete genome sequencing), from different locations of the country, and exhibiting low Ct values for efficient PCR amplification. This study was approved by the Human Bioethical Committee of IVIC.

Total RNA was extracted from nasopharyngeal or nasal swabs samples using the QIAamp Viral RNA Mini Kit (Hilden, Germany). Nested RT-PCR was carried out using the SuperScript III One-Step RT-PCR Platinum Taq HiFi System (Invitrogen, Thermo Fisher Scientific, USA) with specific Artic primers [6] to produce 3 cDNA/tube in six reactions. In some cases cDNA was generated using SuperScript IV VILO Master Mix with ezDNAase (Invitrogen, Thermo Fisher Scientific, Lithuania). The first and second rounds of PCR were performed using high-fidelity *Taq* platinum DNA polymerase (Invitrogen, Thermo Fisher Scientific, USA) to generate overlapping amplicons, of approximately 2000 nt and 1000 nt respectively. These amplicons cover the entire genome. PCR purified fragments were sent to Macrogen Sequencing Service (Macrogen, Korea) for sequencing. Both strands of DNA were sequenced.

Nucleotide sequence data have been deposited into the GenBank database under the accession numbers MT907515-MT907521, MW015946-MW015954 and MW040500-MW040503.

### Phylogenetic analysis

Sequence alignment was performed by the MUSCLE algorithm and the phylogenetic analysis by the Maximum Likelihood method (1000 bootstrap replicas) with MEGA7 [7]. The best-fitting nucleotide substitution model was chosen according to the Akaike Information Criterion (AIC). The phylogenetic inference was also performed with the Bayesian criterion (TN93). The results were similar, but the tree generated with the GTR+G model exhibited better bootstrap values. The evolutionary history was inferred by using the Maximum Likelihood method and the General Time Reversible model. The tree with the highest log likelihood (-42092.99) is shown. The percentage of trees in which the associated taxa clustered together is shown next to the branches. Initial tree(s) for the heuristic search were obtained automatically by applying Neighbor-Join and BioNJ algorithms to a matrix of pairwise distances estimated using the Maximum Composite Likelihood (MCL) approach, and then selecting the topology with superior log likelihood value. A discrete Gamma distribution was used to model evolutionary rate differences among sites (3 categories (+G, parameter = 0.05000)). The tree is drawn to scale, with branch lengths measured in the number of substitutions per site. This analysis involved 42 nucleotide sequences. Codon positions included were 1st+2nd+3rd+Noncoding. All positions containing gaps and missing data were eliminated (complete deletion option). There were a total of 29723 positions in the final dataset. Reference sequences from different lineages [8] were included in the phylogenetic analysis and lineages assigned with the Pangolin COVID-19 Lineage Assigner (https://pangolin.cog-uk.io/).

### Protein structure modelling

NSP1 protein sequences from NCO45512_Wuhan1, MT344955_2 aa del, and 3NE_3 aa del were modelled at SWISS-MODEL servers. NSP1 structures were modeled based on the 7K3N Crystal Structure of NSP1 from SARS-CoV-2 from PDB database as a template used to

generate homology structures. Homology structures were analyzed and superimposed by using UCSF Chimera (*ver.* 1.4).

## Results

A total of 7 full genome sequences were obtained from different regions of Venezuela, mainly from the beginning of the epidemic (Figs 1 and 2). One isolate (the oldest studied one, collected on March 16, 2020) belonged to lineage A.2 (bearing a D in position 614 in the spike). The other 6 isolates belonged to lineage B: two B.1, two B1.1, one B.1.117 and one B.1.5, all with the mutation D614G (Figs 2 and 3). Other 4 isolates from Maracaibo city (Zulia State), for which a 1000 nt sequence surrounding position 23403 was available, also bear the mutation

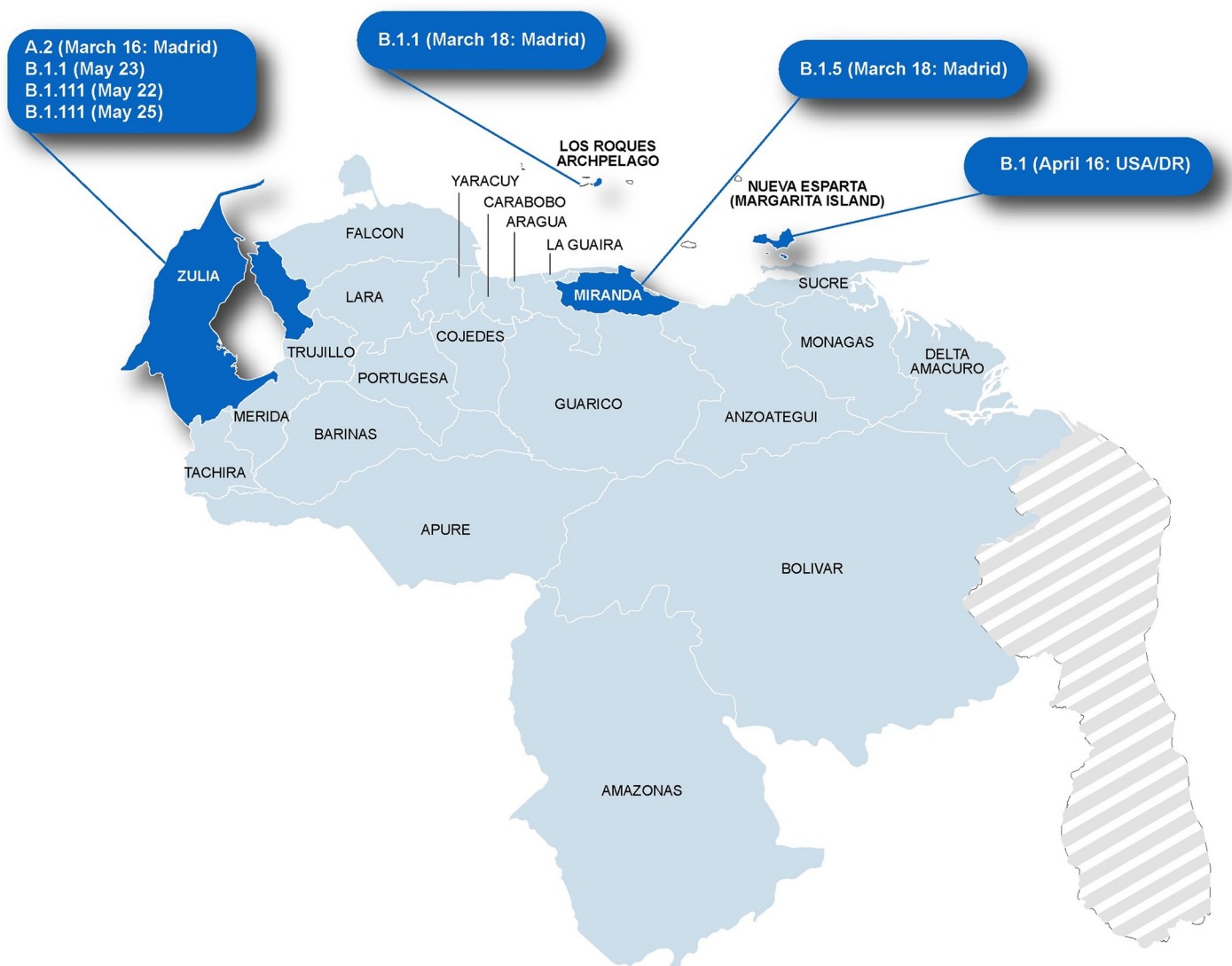

**Fig 2. SARS-CoV-2 Venezuelan isolate lineages according to the place of collection.** The cases are referred to as the date of collection. The country of provenance is indicated for the cases with a recent history of travel and who got probably infected in this country. In the case of Margarita Island, the cases were due to contact with a person coming from the USA and who spent some days in the Dominican Republic (DR). Map of Venezuela was edited and modified from https://freevectormaps.com.

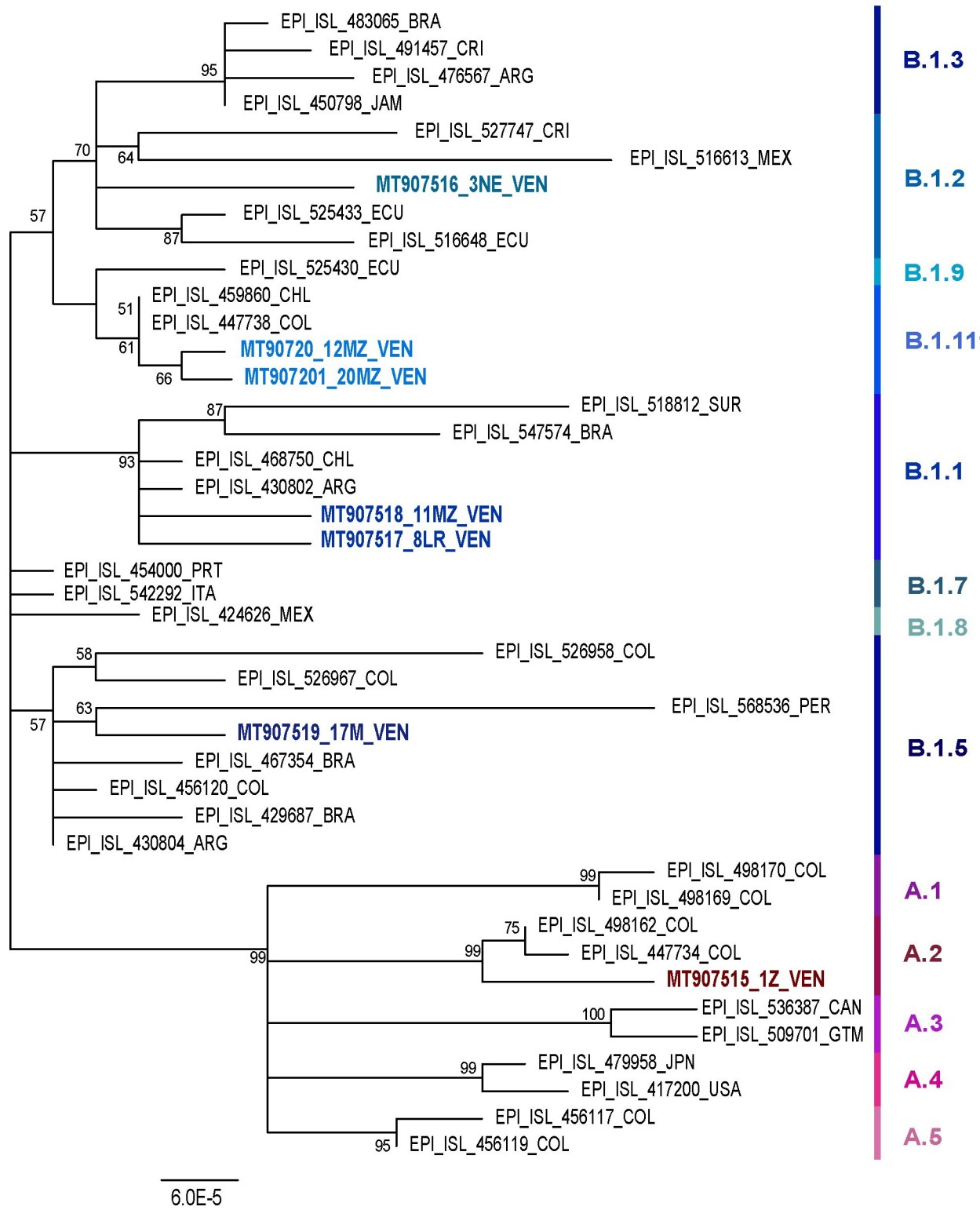

**Fig 3. Phylogenetic tree of Venezuelan complete genome isolates.** The evolutionary history was inferred by using the Maximum Likelihood method (500 bootstrap replicas) and the General Time Reversible model. Sequences are shown by their GenBank accession number or GISAID Initiative (https://www.gisaid.org) identifier, and country of origin. Venezuelan samples are shown in colors with their isolate name. Lineages are shown in different colors. CRI: Costa Rica, MEX: Mexico, ECU: Ecuador, VEN: Venezuela, ARG: Argentina, BRA: Brazil, JAM: Jamaica, CHL: Chile, COL: Colombia, PER: Peru, CAN: Canada, GTM: Guatemala, JPN: Japan, USA: United States of America, ITA: Italy, SUR: Surinam, PRT: Portugal.

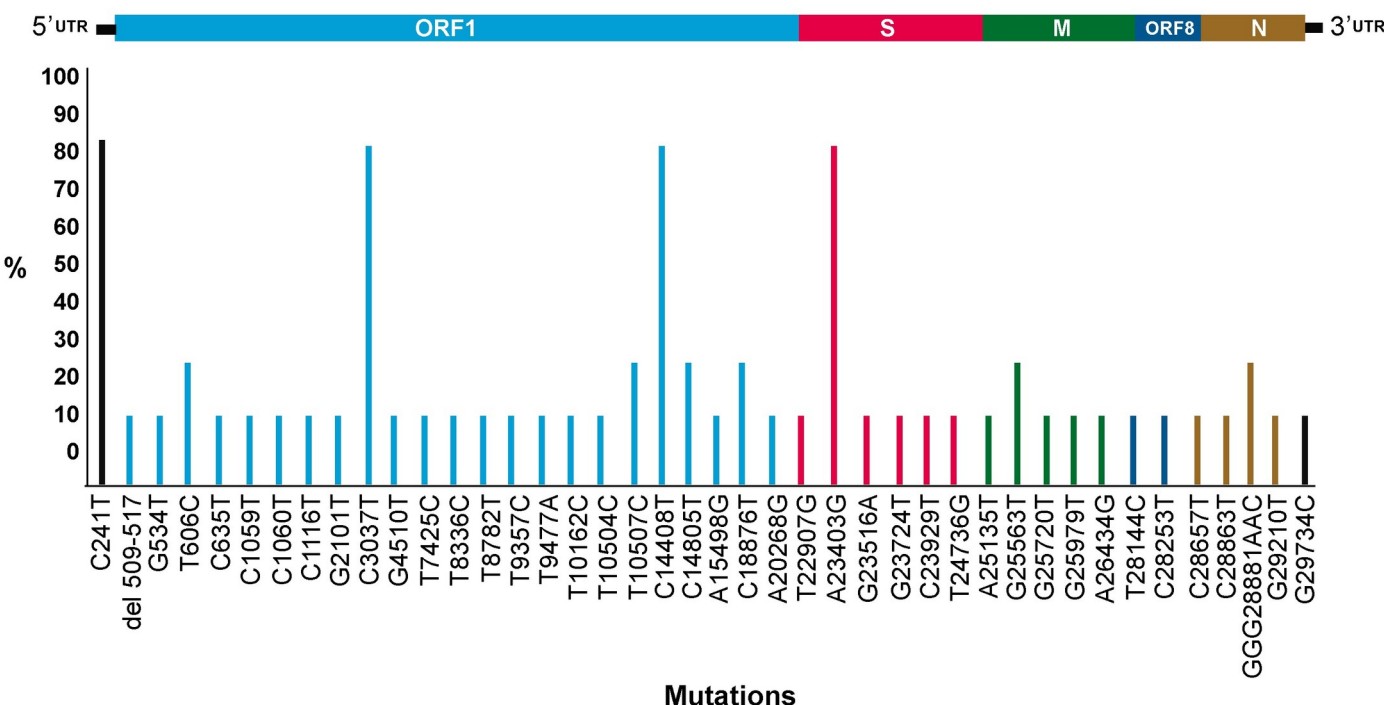

**Fig 4. Frequency of mutations found in Venezuelan SARS-CoV-2 isolates compared to Wuhan MN908947 original isolate.** The position of the mutation in the viral genome is shown.

D614G (Accession numbers MW040500-MW040503), for a total of 10/11 D614G isolates. The samples obtained at the beginning of the epidemic (March 2020) were from individuals returning from Spain, except the 3NE sample collected in Margarita Island (Nueva Esparta State). In contrast, the other samples from Maracaibo were collected in May 2020 and were even community cases or from migrants from Colombia (Fig 2). These samples were indeed related to isolates from Colombia (Fig 3).

Each sample harbored a distinct mutation pattern along the whole genome and exhibited a different sequence (Figs 3 and 4). In addition to the D614G mutation in the spike protein characteristic of lineage B (A23403G), all the Venezuelan sequences belonging to lineage B bore the mutation C241T in the UTR, C14408T in the RdRP, and the synonymous mutation C3037T (Fig 4).

Isolate CoV3NE harbors an in-frame deletion in its sequence, lacking amino acids 83–85 of the NSP1 of the ORF1. Another sample from the same outbreak also exhibited the deletion in its sequence. Similar deletions in this region have been reported in sequences from USA (Fig 5B). The March outbreak of COVID-19 in Margarita Island (Fig 5A) had its origin among a group of young baseball players from Venezuelan´s from League A, who had contact with an American baseball scout who came from Philadelphia and traveled to the Dominican Republic before arriving at the Venezuelan island.

Mercatelli *et al.* [9] described the mutations present in 48,635 sequences available in GISAID Initiative database. We analyzed the number of isolates harboring deletions in these sequences, and found that 1148 sequences out of these 48,635 exhibited at least one deletion. A total of 5,177 deletions were present in these 1,148 sequences, implying an average of 4.5 deletions in these sequences. More than half of these deletions (2,834) were present in the NSP1

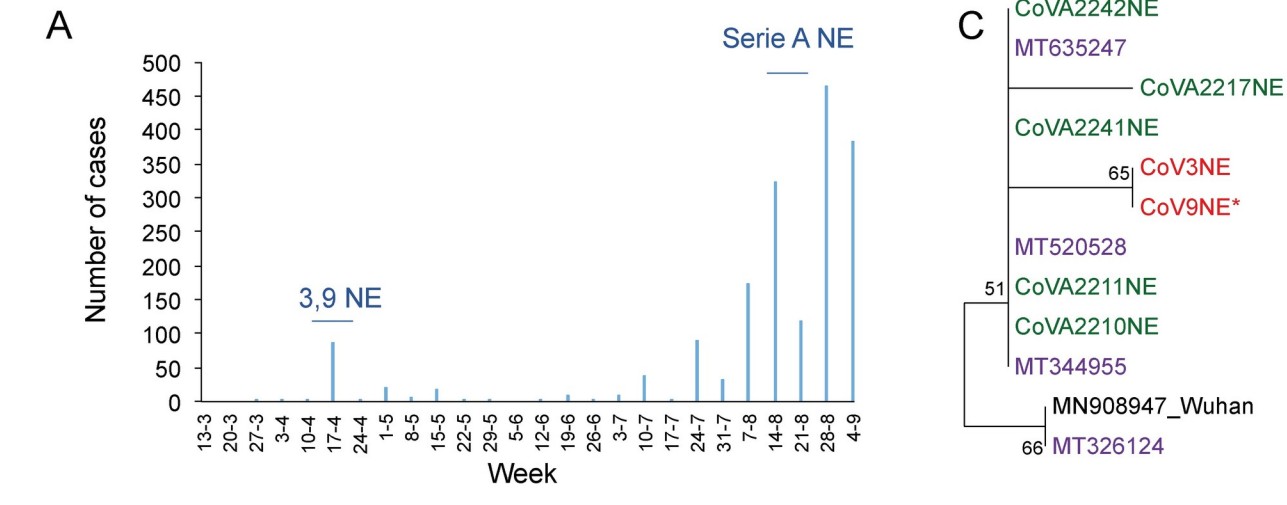

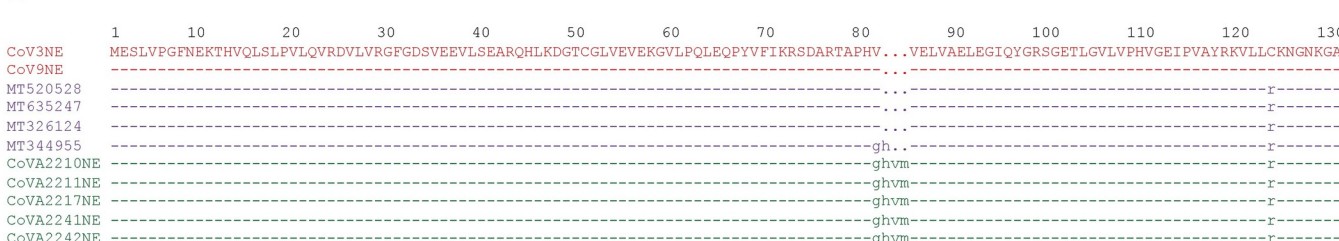

**Fig 5. Analysis of Margarita Island´s outbreak, Nueva Esparta State, Venezuela. A:** COVID-19 cases in Nueva Esparta, according to official data. The cases are referred to the as the date of the report and were downloaded from the web page http://www.ucv.ve/cendes. The horizontal lines indicate the period of time when the samples analyzed were collected. **B:** Phylogenetic tree of the N-terminal region of SARS-CoV-2 genome (nt 40–1946). Genetic distance was estimated by observed divergence, and the phylogenetic tree was constructed with the neighbor-joining method. Samples from Margarita Island (NE) are in red (isolates with deletion) and pink (isolates without deletion), as in C. Samples in blue are from other regions of Venezuela. *Only 812 nt were available for this sequence. **C:** Amino acid alignment of the N-terminal region of the SARS-CoV-2 NSP1 protein, showing the deletion in NE Venezuelan isolates. Other isolates are named according to their GenBank accession number.

protein (Fig 6). Deletions in the region comprising amino acids 82–85 of NSP1, similar to the one found in this study (Fig 5C), were the most frequent among these sequences.

As noted in Fig 5A, COVID-19 epidemics increased again in Margarita Island since July 2020. In order to evaluate if the strain with the deletion in NSP1 was still the responsible for the new cases, the first 1,900 nt were sequenced in isolates from August. The sequence of 1,900 nt from the 5´ end of the genome of 5 isolates was obtained. None of them carry the deletion, and their sequences were different to the one found in the isolates originally circulating in the Island (Fig 5B and 5C). The 5 sequences exhibit at least one nt difference in these 1900 nt, suggesting multiple viral introductions to the Island after July. Other sequences exhibiting a deletion of 2 or 3 amino-acids at the same position were available in the GenBank. These sequences were not related however, to the one found in the isolates from Margarita Island (Fig 5B and 5C), suggesting independent deletion events in these isolates.

The structure of the SARS-CoV-2 NSP1 protein with the 83–85 aa deletion (3NE_3 aa del) was modeled by structure homology in SWISS-MODEL. Structure comparison between 3NE_3 aa del with the NSP1 protein of the first reported isolate of SARS-CoV-2 (NCO45512_-Wuhan1) and the NSP1 protein structure with 84–85 aa deletion (MT344955_2 aa del) showed a structural change (Fig 7A). However, the superimposed NSP1 protein structures (Fig 7B)

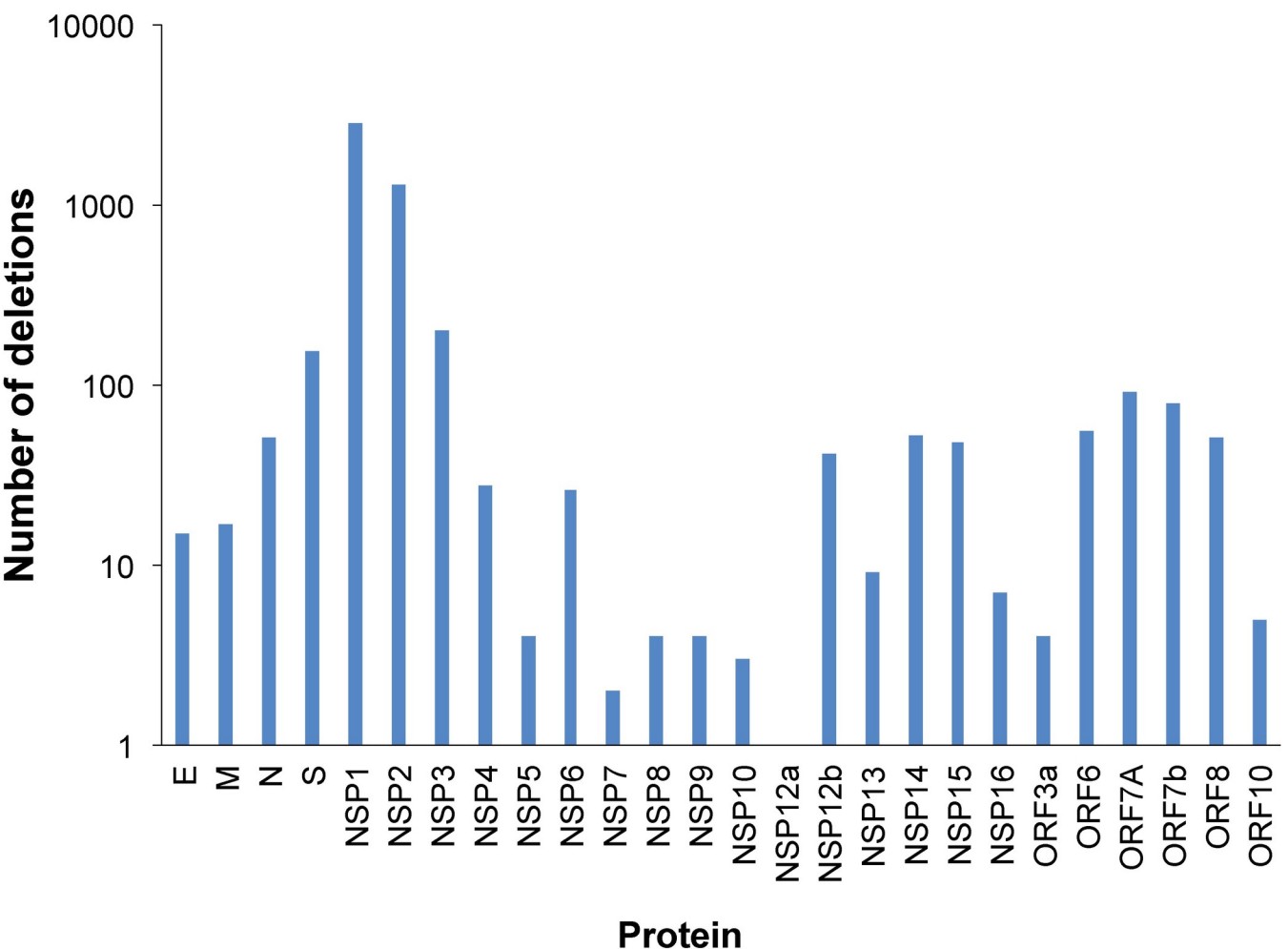

**Fig 6. Analysis of deletions in 48,635 sequences of SARS-CoV-2 isolates.** Number of sequences (out of the 48,635 analyzed by Mercatelli et al. [9]) harboring deletions in a viral protein.

show that these mutations do not affect the core of the protein (elements of the secondary helical structures) and may indicate that this deletion is located in a flexible region.

## Discussion

High diversity was found among the Venezuelan viral isolates sequenced from the early period of the Venezuelan epidemic. The sequences were similar to the one found in other countries of Latin America, with the predominance of lineage B isolates [11–14].

All the lineage B Venezuelan sequences bore the mutation C14408T, which produces the amino acid mutation P314L in the Nsp12 protein, the polymerase in the RdRP. The genomes bearing this mutation were initially associated with significantly higher number of mutations compared to the non-mutated ones [15]. The T14408 is now the most common nucleotide found in the majority of SARS-CoV-2 isolates around the world [16]. As previously described [9], C/T was the most frequent substitution in these complete genome sequences (Table 1). The most probable explanation for this relative abundance is that these genomes are subjected to APOBEC and/or ADAR deaminase activity [17, 18].

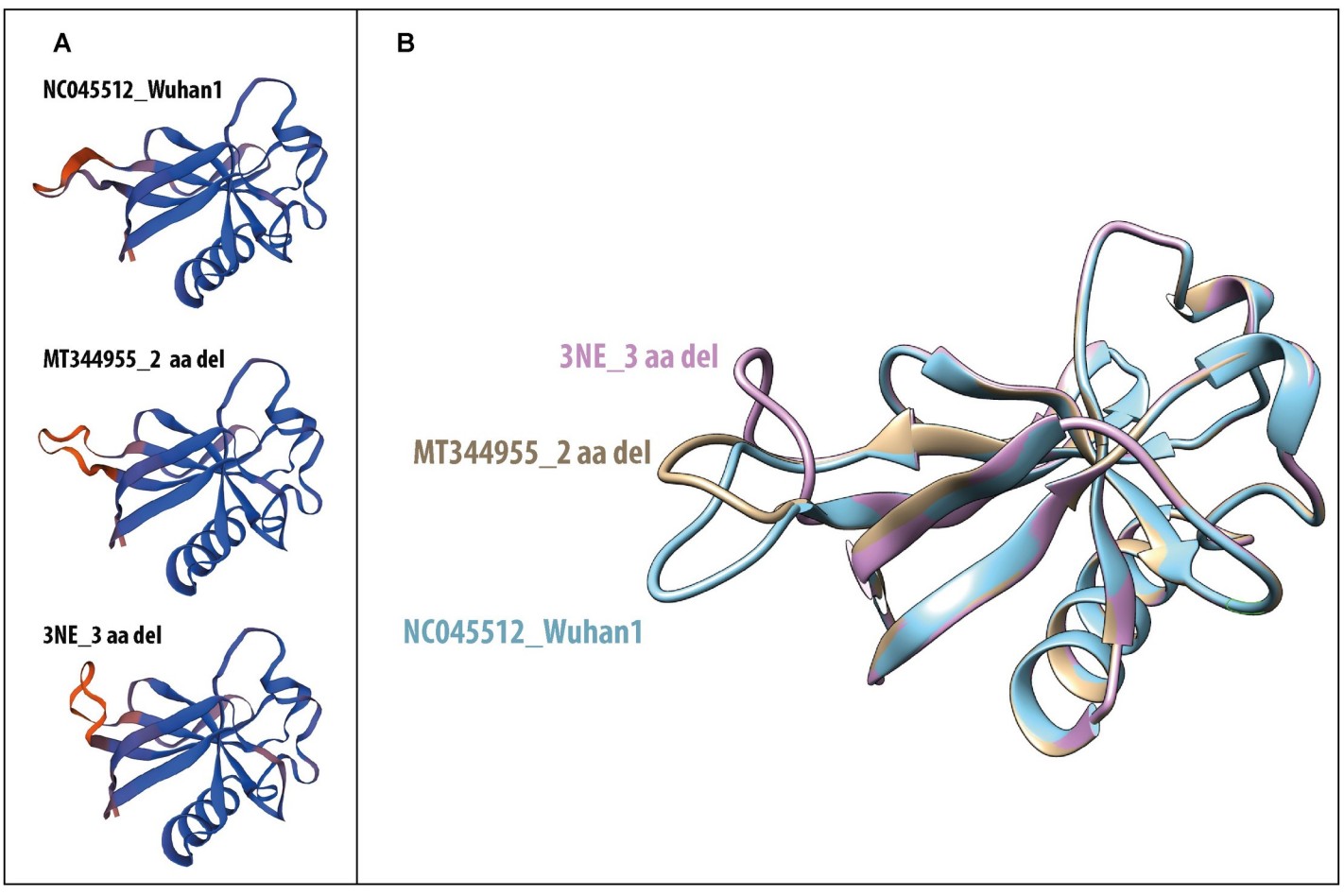

**Fig 7. Modelling of SARS-CoV-2 NSP1 protein with and without deletions. A:** Comparison of complete NSP1 protein structures between the NSP1 protein structure from the first SARS-CoV-2 isolate reported (NCO45512_Wuhan1), and NSP1 protein structure with two amino acid residues deleted (MT344955_2 aa del), and NSP1 protein structure reported in this study with three amino acids deleted (3NE_3 aa del). 7K3N PDB Crystal Structure of NSP1 protein from SARS-CoV-2 was used as a template. **B:** Superimposed structures of the NSP1 proteins from NCO45512_Wuhan1 (Cyan), MT344955_2 aa del (Yellow), and 3NE_3 aa del (Pink). Swiss-Model structure homology-modeling servers was used to generate NSP1 SARS-CoV-2 protein structures [10].

Another non-synonymous mutation A23403G, yielding the mutation D614G in the Spike protein, has been widely described. This mutation is not located in the RBD of the Spike protein, the specific region which interacts with the ACE2 receptor. However, *in vitro* studies with pseudotyped viruses expressing the SARS-CoV-2 spike showed that this mutation facilitated the entry of the virus into the cell, probably through favoring the fusion process. It has been suggested that this variant is more easily transmitted, which might account for its present predominance in the new epidemic foci [11, 19]. This variant has also displaced the original one in new infections in many parts of the world [11]. The higher transmissibility nature of the isolates carrying this mutation is not completely demonstrated, however, and there is no evidence of higher severity associated with this variant [20].

Several deletions have been described in SARS-CoV-2, probably due to the recombinogenic capacity of this viral family [9, 21–24]. As shown in this study, deletions were significantly more frequent in the NSP1 protein, and particularly in the region deleted in the Venezuelan isolates from Margarita Island. NSP1 is the first protein coded at the 5´UTR region of the SARS-CoV-2. This protein is a virulence factor, which plays a role in suppressing host innate

immune functions by inhibiting the expression of type I IFN, inhibiting host protein synthesis, and promoting the degradation of host mRNAs [25, 26]. Some deletions in the NSP1 protein of SARS-CoV have been shown to attenuate this virus. Particularly, a deletion of 10 amino acids in SARS-CoV, comprising the 3 amino acids deleted in the CoV3NE isolate, exhibited a small reduction in viral titer when infecting delayed brain tumor (DBT) mouse cells expressing the murine ACE2 receptor. Nevertheless, attenuation of the viral strain was not observed with this deletion, but with others nearest the C-terminal region of the protein [27]. No fatalities were observed in this first outbreak in Margarita Island. However, it is important to consider that many of the infected persons were young. Structural analysis in NSP1 protein showed a change in the N-terminal domain with the presence of 2 or 3 amino-acid deletions. Both the 2 and 3 amino-acid deletions seemed to induce a structural change in a flexible region of the NSP1 protein. This region has been shown not to play an important role of cellular protein translation [28], but has been suggested to play a role in the regulation of cellular mRNA stability or in suppressing host innate immune functions. More studies are needed to evaluate if the deletion found in the CoV3NE isolate affects the replication capacity of this virus. Another deletion in NSP1 protein has been found circulating in some countries in Europe, the USA, and Brazil [24]. This deletion is located in the C-terminal region of the NSP1 protein, and may affect the activity of this protein in the host´s gene expression regulation.

The largest deletion described so far in the SARS-CoV-2 genome is located in the ORF8. Since a strong immune response has been observed against this protein, this deletion was originally associated with immune evasion [29]. The function of ORF8 is not completely clear. This protein may be associated with immune evasion according to preliminary evidence. In Singapore, although patients infected with viruses carrying this deletion developed clinically significant illness, infections tended to be milder compared with those caused by the wild-type virus, with less pronounced cytokine release during the acute phase of infection [30].

As expected, high diversity was observed between the different isolates from Maracaibo city, since a significant number of introductions shaped the ongoing outbreak in this city. The same seems to occur since July in Margarita Island, after a probable single introduction of the deleted genomic strain in the first wave of the epidemic. This first strain was introduced by air travel to the island, but since then, air connections have been reduced drastically. The apparent diversity of SARS-CoV-2 isolates circulating in this second wave of the epidemic, might have been introduced by contacts with continental Venezuelan inhabitants, traveling by boat to the island.

In conclusion, a high diversity of SARS-CoV-2 isolates was found circulating in Venezuela, with the predominance of the D614G isolates. The first small outbreak in Margarita Island seemed to be associated with a strain carrying a small deletion in the NSP1 protein, but these isolates do not seem to be responsible for the larger outbreak which started in July.

## Author Contributions

**Conceptualization:** Carmen L. Loureiro, Rossana C. Jaspe, Domingo Garzaro, Héctor R. Rangel, Flor H. Pujol.

**Data curation:** Carmen L. Loureiro, Rossana C. Jaspe, José L. Zambrano, Domingo Garzaro, Héctor R. Rangel, Flor H. Pujol.

**Formal analysis:** Carmen L. Loureiro, Rossana C. Jaspe, José L. Zambrano, Héctor R. Rangel, Flor H. Pujol.

**Funding acquisition:** Héctor R. Rangel, Flor H. Pujol.

**Investigation:** Carmen L. Loureiro, Rossana C. Jaspe, Pierina D´Angelo, José L. Zambrano, Lieska Rodriguez, Víctor Alarcon, Mariangel Delgado, Marwan Aguilar, Domingo Garzaro, Flor H. Pujol.

**Methodology:** Carmen L. Loureiro, Rossana C. Jaspe, Pierina D´Angelo, José L. Zambrano, Lieska Rodriguez, Domingo Garzaro, Héctor R. Rangel, Flor H. Pujol.

**Project administration:** Héctor R. Rangel, Flor H. Pujol.

**Resources:** Carmen L. Loureiro, Lieska Rodriguez, Víctor Alarcon, Héctor R. Rangel, Flor H. Pujol.

**Software:** Carmen L. Loureiro, Rossana C. Jaspe, José L. Zambrano, Héctor R. Rangel, Flor H. Pujol.

**Supervision:** Héctor R. Rangel, Flor H. Pujol.

**Validation:** Carmen L. Loureiro, Rossana C. Jaspe, Héctor R. Rangel, Flor H. Pujol.

**Visualization:** Carmen L. Loureiro, Rossana C. Jaspe, Héctor R. Rangel, Flor H. Pujol.

**Writing – original draft:** Flor H. Pujol.

**Writing – review & editing:** Carmen L. Loureiro, Rossana C. Jaspe, Pierina D´Angelo, José L. Zambrano, Lieska Rodriguez, Víctor Alarcon, Mariangel Delgado, Marwan Aguilar, Domingo Garzaro, Héctor R. Rangel, Flor H. Pujol.

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
