## [Decision Letter · Decision Letter 0]

30 Oct 2020

PONE-D-20-31900

SARS-CoV-2 genetic diversity in Venezuela: predominance of D614G variants and analysis of one outbreak

PLOS ONE

Dear Dr. Pujol,

Thank you for submitting your manuscript to PLOS ONE. After careful consideration, we feel that it has merit but does not fully meet PLOS ONE’s publication criteria as it currently stands. Therefore, we invite you to submit a revised version of the manuscript that addresses the points raised during the review process.

We look forward to receiving your revised manuscript.

Kind regards,

Marta Giovanetti, Ph.D.

Academic Editor

PLOS ONE

2. We note that Figure 2 in your submission contain map images which may be copyrighted. All PLOS content is published under the Creative Commons Attribution License (CC BY 4.0), which means that the manuscript, images, and Supporting Information files will be freely available online, and any third party is permitted to access, download, copy, distribute, and use these materials in any way, even commercially, with proper attribution. For these reasons, we cannot publish previously copyrighted maps or satellite images created using proprietary data, such as Google software (Google Maps, Street View, and Earth). For more information, see our copyright guidelines: http://journals.plos.org/plosone/s/licenses-and-copyright.

(a) You may seek permission from the original copyright holder of Figure 2 to publish the content specifically under the CC BY 4.0 license.

(b) If you are unable to obtain permission from the original copyright holder to publish these figures under the CC BY 4.0 license or if the copyright holder’s requirements are incompatible with the CC BY 4.0 license, please either i) remove the figure or ii) supply a replacement figure that complies with the CC BY 4.0 license. Please check copyright information on all replacement figures and update the figure caption with source information. If applicable, please specify in the figure caption text when a figure is similar but not identical to the original image and is therefore for illustrative purposes only.

Reviewers' comments:

Reviewer's Responses to Questions

**Comments to the Author**

1. Is the manuscript technically sound, and do the data support the conclusions?

Reviewer #1: Yes

Reviewer #2: Partly

Reviewer #3: Yes

2. Has the statistical analysis been performed appropriately and rigorously? 

Reviewer #1: Yes

Reviewer #2: No

Reviewer #3: N/A

3. Have the authors made all data underlying the findings in their manuscript fully available?

Reviewer #1: Yes

Reviewer #2: No

Reviewer #3: Yes

4. Is the manuscript presented in an intelligible fashion and written in standard English?

Reviewer #1: Yes

Reviewer #2: No

Reviewer #3: Yes

5. Review Comments to the Author

Reviewer #1: "The Prevalence and Hearing Outcome of Congenital CMV Infection in an Urban Area of East Asia: Population-based Study" by Yang et al. is well-written and provides a relatively comprehensive view about Congenital cytomegalovirus prevalence in Taipei, Taiwan. Although the study has limitations already recognized by the authors, this manuscript provide advances in our understanding about the relevance the early identification of cCMV in neonates increasing the quality of life of patients and better follow-up of the public service to these patients telling a nice story. I think it would be suitable for publication in PLoS One after careful revision of English language.

Reviewer #2: - Highlights sentences do not reflect the content of this article.

- Today there are many Sars-CoV-2 genomes in different databases. We still need to insert a larger number of genomes for this type of analysis.

- Complementary methodologies are required for phylogenetic and phylogenomics analysis.

- Explain the parameters that have been selected for the models used in this article.

- Provide the criteria used for the analysis of these variants.

- The data presented do not support the conclusion suggested by the authors.

- Improve the image resolution in all figures.

Reviewer #3: In this study, the authors analyzed full genomes sequences obtained from different regions of Venezuela. They found that the majority of the isolates belongs to lineage B and they bear the mutation D614G in the Spike protein. They also found an in-frame deletion (amino acids 83-85) of the nsp1.

- The authors mentioned both in the abstract and in the results section that nsp1 exhibits high frequency of deletions. The authors should add the following references of recent studies that showed deletions in nsp1 of SARS-CoV-2:

o F. Benedetti, G.A. Snyder, M. Giovanetti, S. Angeletti, R.C. Gallo, M. Ciccozzi, D. Zella, Emerging of a SARS-CoV-2 viral strain with a deletion in nsp1, Journal of Translational Medicine 18, (11) (2020) 3-29.

o T. Phan, Genetic diversity and evolution of SARS-CoV-2, Infect Genet Evol 81, (11) (2020)104- 260.

o M.R. Islam, Genome-wide analysis of SARS-CoV-2 virus strains circulating worldwide implicates heterogeneity, Scientific Reports 10, (12) (2020) 14-40.

- Does the deletion in nsp1 analyzed by the authors correlate with potential decreased viral pathogenicity? Does it impact the core structure of nsp1? The authors should better clarify how this deletion can impact the functionality and the pathogenesis of SARS-CoV-2.

- What are the percentage of mutations? The authors should revise Table 1 and add the number of sequences carrying the mutations/deletions and the related percentages to make the table better informative.

- A few typos need to be corrected

6. PLOS authors have the option to publish the peer review history of their article (what does this mean?). If published, this will include your full peer review and any attached files.

Reviewer #1: No

Reviewer #2: No

Reviewer #3: No

---

## [Author Response · Author response to Decision Letter 0]

27 Nov 2020

PONE-D-20-31900

SARS-CoV-2 genetic diversity in Venezuela: predominance of D614G variants and analysis of one outbreak

https://journals.plos.org/plosone/s/file?id=wjVg/PLOSOne_formatting_sample_main_body.pdf and https://journals.plos.org/plosone/s/file?id=ba62/PLOSOne_formatting_ sample_title_authaut_affiliations.pdf

We have edited our manuscript according to the style requirements.

2. We note that Figure 2 in your submission contain map images which may be copyrighted. All PLOS content is published under the Creative Commons Attribution License (CC BY 4.0), which means that the manuscript, images, and Supporting Information files will be freely available online, and any third party is permitted to access, download, copy, distribute, and use these materials in any way, even commercially, with proper attribution. For these reasons, we cannot publish previously copyrighted maps or satellite images created using proprietary data, such as Google software (Google Maps, Street View, and Earth). For more information, see our copyright guidelines: http://journals.plos.org/plosone/s/licenses-and-copyright.

The map was edited and modified from https://freevectormaps.com, with no copyright. This information is included in Fig 2.

Note: HTML markup is below. Please do not edit.]

There is no more supporting information since the cited paper is now published.

Reviewers' comments:

Reviewer's Responses to Questions

Comments to the Author

1. Is the manuscript technically sound, and do the data support the conclusions?

Reviewer #1: Yes

Reviewer #2: Partly

Reviewer #3: Yes

We modified the text to a less affirmative style because of some limitations of this study (see below), in Abstract and Conclusions.

2. Has the statistical analysis been performed appropriately and rigorously?

Reviewer #1: Yes

Reviewer #2: No

Reviewer #3: N/A

More information was included the model used for phylogenetic analysis (see Materials and Methods, page 6 lines 100-103). There is no other statistical analysis in this study.

3. Have the authors made all data underlying the findings in their manuscript fully available?

Reviewer #1: Yes

Reviewer #2: No

Reviewer #3: Yes

In terms of Data Availability, we submitted previously all our sequences to GenBank. 

4. Is the manuscript presented in an intelligible fashion and written in standard English?

Reviewer #1: Yes

Reviewer #2: No

Reviewer #3: Yes

We have revised the manuscript for English accuracy.

5. Review Comments to the Author

Reviewer #1: "The Prevalence and Hearing Outcome of Congenital CMV Infection in an Urban Area of East Asia: Population-based Study" by Yang et al. is well-written and provides a relatively comprehensive view about Congenital cytomegalovirus prevalence in Taipei, Taiwan. Although the study has limitations already recognized by the authors, this manuscript provide advances in our understanding about the relevance the early identification of cCMV in neonates increasing the quality of life of patients and better follow-up of the public service to these patients telling a nice story. I think it would be suitable for publication in PLoS One after careful revision of English language.

These comments do not refer to our manuscript and then were not addressed.

Reviewer #2: 

- Highlights sentences do not reflect the content of this article.

We do not know exactly which highlights sentences the reviewer is referring to. We have modified the abstract in order to better reflect the content of the article, including the new information of modelling. As stated before, we modified the text to a less affirmative one.

- Today there are many Sars-CoV-2 genomes in different databases. We still need to insert a larger number of genomes for this type of analysis.

We acknowledge the comment of the reviewer and agree that this study involves a relative low number of sequences produced from Venezuelan isolates. However, since the sequences were chosen at the beginning of the epidemic and from different locations, representative of the initial outbreaks occurring in the country, we think that the study is at least a partial reflection of the initial viral diversity found in Venezuela. Indeed, none of the isolates were genetically identical, supporting this hypothesis. The manuscript also includes the analysis of 48.635 SARS-CoV-2 sequences available in GISAID data base and described previously by other authors, but we analyzed in detail the frequency of deletions in these sequences. A new figure is added to better describe this analysis (Fig 6). In addition, the revised version includes a more in-deep analysis of the strain exhibiting a deletion in NSP1, with a new figure (Fig 7). 

- Complementary methodologies are required for phylogenetic and phylogenomics analysis.

We acknowledge the comment. Several phylogenetic inferences were performed (see below in parameters), arising basically the same topology for the phylogenetic tress. 

Modelling of the deleted NSP1 protein was included in this new version and discussed (Figure 7, Abstract, end of Results in page 15 and Discussion in page 18).

- Explain the parameters that have been selected for the models used in this article.

The parameters used for the models for phylogenetic inference were described in Materials and Methods (Page 6). A short description was added in the legend of Figure 3 (Page 10). In our work, the phylogenetic tree was constructed based on the GTR+G model according to the Akaike criterion. The phylogenetic inference was also performed with the Bayesian criterion (TN93). The results were similar, but the tree generated with the GTR+G model exhibited better bootstrap values.

- Provide the criteria used for the analysis of these variants.

The criteria of selection of samples were added in Page 5, lines 79-81: samples from different part of the country and from the beginning of the epidemic, with low Ct value to guarantee efficient amplification. This information was included in the first paragraph of Page 5.

- The data presented do not support the conclusion suggested by the authors.

As sated before, we have modified the conclusions to a less affirmative style.

- Improve the image resolution in all figures.

Image resolution of figures was improved.

Reviewer #3: In this study, the authors analyzed full genomes sequences obtained from different regions of Venezuela. They found that the majority of the isolates belongs to lineage B and they bear the mutation D614G in the Spike protein. They also found an in-frame deletion (amino acids 83-85) of the nsp1.

- The authors mentioned both in the abstract and in the results section that nsp1 exhibits high frequency of deletions. The authors should add the following references of recent studies that showed deletions in nsp1 of SARS-CoV-2:

o F. Benedetti, G.A. Snyder, M. Giovanetti, S. Angeletti, R.C. Gallo, M. Ciccozzi, D. Zella, Emerging of a SARS-CoV-2 viral strain with a deletion in nsp1, Journal of Translational Medicine 18, (11) (2020) 3-29.

o T. Phan, Genetic diversity and evolution of SARS-CoV-2, Infect Genet Evol 81, (11) (2020)104- 260.

o M.R. Islam, Genome-wide analysis of SARS-CoV-2 virus strains circulating worldwide implicates heterogeneity, Scientific Reports 10, (12) (2020) 14-40.

The references were included and discussed (Page 18, line 271 and Page 19, line 291-292). Thank you very much.

- Does the deletion in nsp1 analyzed by the authors correlate with potential decreased viral pathogenicity? Does it impact the core structure of nsp1? The authors should better clarify how this deletion can impact the functionality and the pathogenesis of SARS-CoV-2.

We wish to thank the reviewer for this comment. We include a modelling analysis of the deleted NSP1 proteins and found indeed some differences compared to the wt protein. Information was included in Abstract, Materials and Methods, Results, Discussion and a Figure 6 was added, as stated before. A new co-author, responsible for this analysis, is also included.

- What are the percentage of mutations? The authors should revise Table 1 and add the number of sequences carrying the mutations/deletions and the related percentages to make the table better informative.

The Table was substituted by a figure which is more informative (Fig 4). 

- A few typos need to be corrected.

The manuscript was corrected for English accuracy.

---

## [Decision Letter · Decision Letter 1]

19 Jan 2021

PONE-D-20-31900R1

SARS-CoV-2 genetic diversity in Venezuela: predominance of D614G variants and analysis of one outbreak

PLOS ONE

Dear Prof. Pujol,

Thank you for submitting your manuscript to PLOS ONE. After careful consideration, we invite you to submit a revised version of the manuscript that addresses the points raised during the review process.

We look forward to receiving your revised manuscript.

Kind regards,

Ahmed S. Abdel-Moneim, Ph.D.

Academic Editor

PLOS ONE

Reviewers' comments:

Reviewer's Responses to Questions

**Comments to the Author**

1. If the authors have adequately addressed your comments raised in a previous round of review and you feel that this manuscript is now acceptable for publication, you may indicate that here to bypass the “Comments to the Author” section, enter your conflict of interest statement in the “Confidential to Editor” section, and submit your "Accept" recommendation.

Reviewer #2: All comments have been addressed

2. Is the manuscript technically sound, and do the data support the conclusions?

Reviewer #2: Partly

3. Has the statistical analysis been performed appropriately and rigorously? 

Reviewer #2: N/A

4. Have the authors made all data underlying the findings in their manuscript fully available?

Reviewer #2: Yes

5. Is the manuscript presented in an intelligible fashion and written in standard English?

Reviewer #2: Yes

6. Review Comments to the Author

Reviewer #2: We noticed that the authors responded to the reviewers' requests.

**Note that there are still two points that need to be revised: increase the number to 1000 bootstrap iterations for better analysis and all Figures need improvement in resolution and colors, in particular Figure 7.**

**The Figure 6 needs to be done with two scales in order to identify the lowest values.**

7. PLOS authors have the option to publish the peer review history of their article (what does this mean?). If published, this will include your full peer review and any attached files.

Reviewer #2: No

---

## [Author Response · Author response to Decision Letter 1]

29 Jan 2021

PONE-D-20-31900R1

SARS-CoV-2 genetic diversity in Venezuela: predominance of D614G variants and analysis of one outbreak

There are still two points that need to be revised: 

increase the number to 1000 bootstrap iterations for better analysis 

Bootstrap iterations were increased to 1000. New Figure is included and the bootstrap replicas was corrected to1000 in Materials and Methods, page 6.

and all Figures need improvement in resolution and colors, in particular Figure 7.

Figures are now included with better resolution and colors were slightly modified in Figure 7, for better appreciation.

The Figure 6 needs to be done with two scales in order to identify the lowest values.

The scale of Figure 6 was changed to a logarithmic one for better evaluation of lowest values.

---

## [Editor Report · Decision Letter 2]

3 Feb 2021

SARS-CoV-2 genetic diversity in Venezuela: predominance of D614G variants and analysis of one outbreak

PONE-D-20-31900R2

Dear Dr. Pujol,

We’re pleased to inform you that your manuscript has been judged scientifically suitable for publication and will be formally accepted for publication once it meets all outstanding technical requirements.

Kind regards,

Ahmed S. Abdel-Moneim, Ph.D.

Academic Editor

PLOS ONE

---

## [Editor Report · Acceptance letter]

8 Feb 2021

PONE-D-20-31900R2 

SARS-CoV-2 genetic diversity in Venezuela: predominance of D614G variants and analysis of one outbreak 

Dear Dr. Pujol:

I'm pleased to inform you that your manuscript has been deemed suitable for publication in PLOS ONE. Congratulations! Your manuscript is now with our production department. 

Kind regards, 

on behalf of

Dr. Ahmed S. Abdel-Moneim 

Academic Editor

PLOS ONE